# DQVis Dataset: Natural Language to Biomedical Visualization

**Devin Lange**
Harvard Medical School
devin@hms.harvard.edu

**Pengwei Sui**
Harvard Medical School
pengwei_sui@hms.harvard.edu

**Shanghua Gao**
Harvard Medical School
shanghua_gao@hms.harvard.edu

**Marinka Zitnik**
Harvard Medical School
marinka@hms.harvard.edu

**Nils Gehlenborg**
Harvard Medical School
nils@hms.harvard.edu

## Abstract

Biomedical research data portals are essential resources for scientific inquiry, and interactive exploratory visualizations are an integral component for querying such data repositories. Increasingly, machine learning is being integrated into visualization systems to create natural language interfaces where questions about data can be answered with visualizations, and follow-up questions can build on the previous state. This paper introduces a framework that takes abstract low-level questions about data and a visualization grammar specification that can answer such a question, reifies them with data entities and fields that meet certain constraints, and paraphrases the question language to produce the final collection of realized data-question-visualization triplets. Furthermore, we can link these foundational elements together to construct chains of queries, visualizations, and follow-up queries. We developed an open-source review interface for evaluating the results of these datasets. We applied this framework to five biomedical research data repositories, resulting in DQVis, a dataset of 1.08 million data-question-visualization triplets and 11.4 thousand two-step question samples. Five visualization experts provided feedback on the generated dataset through our review interface. We present a summary of their input and publish the full reviews as an additional resource alongside the dataset. The DQVis dataset and generation code are available at `https://huggingface.co/datasets/HIDIVE/DQVis` and `https://github.com/hms-dbmi/DQVis-Generation`.

## 1 Introduction

Natural language interfaces show promise for exploratory data analysis [42]. The central engine of such a system converts a natural language query into a visualization. There are different approaches to this engine, but many require training data in the form of natural language queries and visualization responses (NL2VIS). With modern techniques, such datasets carry various requirements. First, they must be large enough to fine-tune an LLM. Second, many applications are tailored towards specific domains, often requiring the questions and visualizations to be rooted in this domain. Finally, the goal of NLI is not to answer a single question with a visualization, but to provide an interface for exploring data, where the past questions and visualizations are known by the NL2VIS generation

39th Conference on Neural Information Processing Systems (NeurIPS 2025) Track on Datasets and Benchmarks.

engine, in other words, multi-step reasoning. All of these requirements for sufficient training data coalesce into a challenging, labor-intensive task.

Such training data is worthwhile because of the potential value of NLI. In biomedical research, large research consortia typically build and curate data repositories to collect and distribute data generated in the course of collaborative multiyear projects. Researchers exploring these repositories can lead to discoveries explaining the mechanisms of human biology and the development of new treatments for diseases like cancer [3, 14, 8]. Many biomedical research data repositories contain sophisticated interfaces on their data portal for interacting with the data[44, 10, 37]. However, such interfaces require time and expertise to develop, and it is difficult to account for every possible way a user may want to query the data and design visualizations and interfaces that can accommodate all of these. Here is the promise of natural language interfaces. An ideal system can bridge the gap between diverse user queries and one-size-fits-all interfaces by responding to each individual with the response to the exact question they have, whether that's a developer trying to understand the size and type of datasets on the repository, or a researcher seeking the next critical piece of information in the path to solve cancer.

Building an ideal NLI interface is challenging with insufficient training data. Unfortunately, the construction of datasets requires time, expertise, and computational resources. LLMs can be fragile when dealing with terms in evolving specialized domains [13], such as new drug names or biological assays. Thus, any curation of a domain-specific training dataset will require expertise in the construction and review of the data. The time required for this construction is exacerbated by the large scale of data required — experts can't construct all the required data manually. Therefore, computational resources are often employed to assist in the creation of these large-scale datasets. Finally, supporting multi-step further complicates the task by introducing branching complexity of possible paths.

NVBench [30] is an existing dataset over 25K data points and has already shown utility in training NL2VIS models [31]. However, as a domain-agnostic dataset, it may not be appropriate for domain-specific applications. Additionally, although the dataset size was sufficient for techniques a few years ago, it may not scale to popular modern-day approaches, such as fine-tuning LLMs. Finally, NVBench does not consider multi-step reasoning.

We work towards addressing these challenges with the **two main contributions of this paper**, a data-question-visualization generation framework, and a new dataset (DQVis) for biomedical research repositories constructed with this framework. DQVis includes **1.08 million data-question-visualization triplets** that address the challenge of domain specificity and data scale for this particular domain. Additionally, we provide **11.4K two-step question samples** that are representative of common visual exploration tasks. The framework we introduce eases the burden for other domains to create datasets by taking as input a list of dataset schemas and a relatively small number of abstract questions, reifying those questions with the provided data, and paraphrasing them to produce a massive number of resulting data-question-visualization data points. We also include an interface for reviewing generated data as part of this framework. Finally, we illustrate how data from DQVis can be linked together, forming multi-step chains of user queries and visualization responses.

## 2 Related Work

### 2.1 Natural Language to Data Visualization

Data visualization is a ubiquitous way for people to interact with data. However, selecting a visualization manually for a given dataset is tedious. Thus, the visualization community has long researched visualization recommendation systems that can automatically produce visualizations for a given dataset or integrate AI into visualization systems [34, 54, 12, 36, 55]. Taking it further, natural language interfaces introduce a system that allows users to produce or update visualizations [42]. This technique can be used to generate visualizations given a prompt [6, 35, 31], to update interactive visualization systems [41, 19, 21, 57], and to support visualization authoring systems [52, 43, 32]. In the last few years, the use of LLMs for such systems has exploded [32, 28, 50, 26, 59, 29, 43, 56, 49, 24, 11]. In addition, work has been done to understand what the visualization design preferences of LLMs are [51], their promises and pitfalls [7], and how to evaluate them [5]. Many of these systems are designed for a specific domain, such as health information seeking [56], education [15], and sports [26, 29, 59]. Furthermore, many are tailored towards a specific data type, like graphs [45] or

trajectories [21]. Although some systems are able to use standard LLMs, others fine-tune the LLM [16, 15], highlighting the need for domain-specific datasets that can readily be used for this task.

Natural-language-to-visualization (NL2VIS) requires models to analyze and transform user queries expressed in natural language into visualizations. NVBench [30] repurposed Spider [58] to generate Vega-Lite specifications from text, focusing on the general domain. NVBench is an important contributions to the field, however, DQVis covers additional challenges related to scale, domain, and multi-step queries. NVBench includes 25,750 datapoints, whereas DQVis includes 1,075,190 datapoints, a more than 40x increase in scale. DQVis is a domain-specific dataset centered around biomedical research repositories, whereas NVBench is a general-purpose dataset. Finally, DQVis contains multi-step data, which is increasingly vital for conversational LLMs. NVBench does not include multi-step data. Dial-NVBench builds on the NVBench dataset to include dialog-style constructions of data visualizations [46], however, it does not cover address the challenge of scale and domain. VizNet [20] assembled a large repository of real-world examples across general datasets but did not consider natural language input. Srinivasan et al. [47] collected specification language utterances for describing data visualizations. AVA [28] introduced iterative refinement of visualizations using multimodal agents, although this was limited to narrow domains. While these works focus on general-purpose visualization datasets, they do not address biomedical research data, which requires an understanding of domain-specific knowledge. In contrast, our dataset is specifically designed for the biomedical domain. Furthermore, we account for the complexity of biomedical visualization queries, which often demand detailed and multi-step reasoning to produce accurate and meaningful visualizations.

## 2.2 Data Visualization Question Answering

Data visualization question answering (DVQA), inspired by visual question answering (VQA) aims to train models to answer questions about visualizations [18, 9]. The goal of DVQA is similar to our aim of answering questions about data *with* visualizations, but also has critical distinctions. The high-level approach of creating datasets [23, 22, 2] in order to train models [2] is similar to our goals. Although some data retrieval questions could exist in both DVQA and DQVis datasets, e.g. "What category contains the most records?" could be asked of a dataset, and a bar chart of category record counts, the *best* visualization for answering this retrieval question may not be the same as a visualization that could contain the answer. Additionally, some questions should exist in an NL2VIS dataset, but not in a DVQA dataset. One example is questions holistically characterizing data, e.g. "What is the distribution of values" can be answered with a histogram, but asking "What is the distribution of values" for a histogram is not an appropriate question. Some questions can exist in DVQA, but not in DQVis. In particular, questions about the structure of a chart "Is the y-axis scale linear?" fall into this category [22]. Furthermore, while DVQA datasets can omit the dataset the data visualization is representing, DQVis must include the data since this is the artifact on which the questions are posed. Still, some aspects can be applied to both scenarios. Ko et al. introduce a framework for generating DVQA datasets, and we use a similar technique for paraphrasing in our framework, inspired by their work [25]. In short, DVQA and DQVis are both important areas with overlap, but one cannot simply reverse a dataset in one domain to produce the other.

## 3 Domain Background

The framework we developed is designed to support creating domain-specific datasets. In this paper, we illustrate its utility for biomedical research data portal metadata. The details of data portals vary, but a common theme is that they include metadata on **donors** (e.g., humans or mice) who provide biological **samples** (e.g., blood or tissue), which are analyzed and result in a **dataset**. We distinguish the metadata from the datasets themselves and focus on metadata in this work. Whereas the datasets contain the results of various biological assays, the metadata records information about the datasets, e.g., which assay was run and how large the dataset is. Datasets are essential for deep analysis of an individual sample, and metadata is needed to identify relevant datasets or understand patterns across multiple samples.

We use an entity relationship model to represent the metadata. **Entities** (E) correspond to a data table, such as donors. The **fields** (F) correspond to columns in the data table that represent attributes of the entity, such as weight or height. Fields can have different data types, such as *quantitative*, *ordinal*,

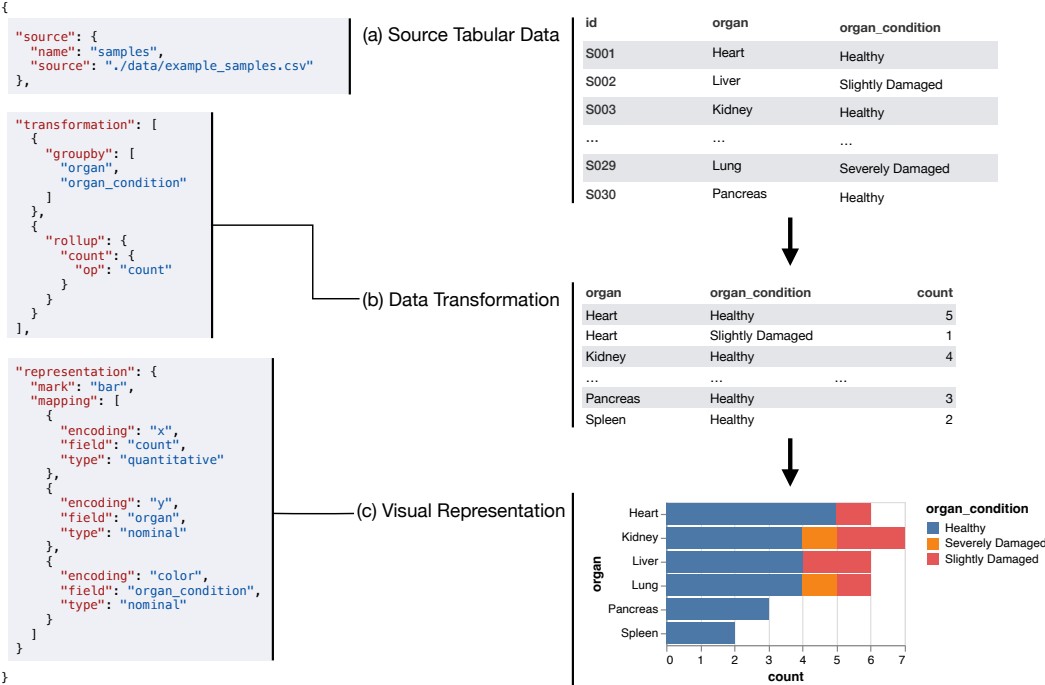

Figure 1: The biomedical visualization grammar we are developing has three main components: (a) specifying one or more sources of tabular data; (b) transforming that data by filtering, grouping, and aggregating; (c) mapping transformed data to visual channels.

and *nominal*. Entities can also be related through identifying columns as one-to-one, one-to-many, or many-to-many. For instance, you might expect a biological sample to have a one-to-one or many-to-one relationship with donors.

There are many tools and libraries for visualizing such data. In our case, we focus on a visualization grammar we are developing for biomedical data portal exploration.[1] This grammar is defined by a JSON Schema, which defines how visualizations can be constructed with a declarative JSON specification (spec). Our visualization toolkit renders these specifications into interactive visualizations (see Figure 1).[2] At a high level, this specification will define one or more tabular sources of data (e.g., CSV files), how the data is transformed (e.g., grouped and aggregated), and how the transformed data is mapped to visual channels (e.g., color and position). This grammar of graphics approach is popular among the visualization community [53, 39, 33]. In particular, our grammar is similar to a popular library, vega-lite [39], and uses it for part of the rendering implementation. The most significant difference is that our grammar includes additional support for tabular representations (see Figure 2e), an essential component of biomedical research data portals.

## 4 Data Description

The core component of the DQVis dataset is triplets of **data** (D), **query** (Q) that could be posed about such data, and **visualizations** (Vis) that could answer those questions. The **data** column references an entity relationship definition. The **query** is in the form of a natural language query a user may have about a dataset. Finally, the **visualization** is a JSON specification for our grammar. DQVis contains 1.08 million data-query-visualization triplets across various data repositories and chart complexity, sumarized in Table 1. Examples of generated questions and visualizations is shown in Figure 2.

---

[1]https://hms-dbmi.github.io/udi-grammar

[2]https://github.com/hms-dbmi/udi-grammar

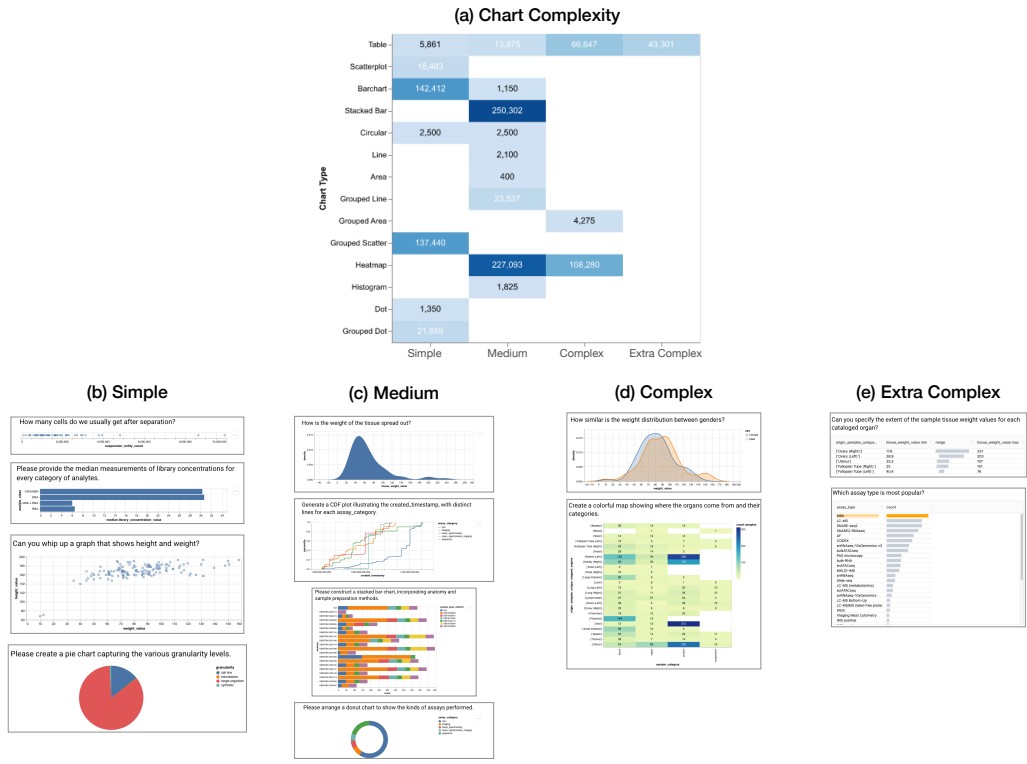

Figure 2: (a) DQVis contains 14 chart types with varying degrees of complexity. Complexity is determined by the number of keys in the visualization specification. (b) Simple visualizations have <= 12 keys; (c) Medium <= 24; (d) Complex <= 36; and (e) Extra Complex is > 36.

| | Dataset Dimensions | | Chart Complexity | | | |
|---|---|---|---|---|---|---|
| **Repository** | Entities | Fields | Simple | Medium | Complex | Extra Complex |
| HuBMAP | 3 | 320 | 321,069 | 514,813 | 147,680 | 41,695 |
| MW | 22 | 99 | 3,375 | 3,994 | 10,950 | 681 |
| 4DN | 20 | 101 | 2,661 | 1,800 | 10,225 | 525 |
| MoTrPAC | 14 | 68 | 2,075 | 1,525 | 7,000 | 275 |
| SenNet | 6 | 35 | 725 | 650 | 3,347 | 125 |

Table 1: DQVis includes data from 5 different biomedical research data repositories: HuBMAP [44], MW[48], 4DN[10, 37], MoTrPAC[38], and SenNet [27]. The different datasets contain a varied number of Entities and Fields, visualized with different types of chart complexity.

The entity relationship model for the biomedical research $data$ is saved in a Frictionless Data Package.[3] This standard provides a consistent definition for entity relationships and fields within each entity and can be extended with additional information. For our framework, we extend data packages to include other useful information for each field, such as the number of unique values recorded. The Common Fund Data Ecosystem (CFDE [4] publishes metadata packages for multiple data portals in the Crosscut Metadata Model (C2M2) [4] format, which also adheres to the Frictionless Data Package standard. DQVis contains five different data packages from biomedical data portals. SenNet [27], Metabolomics Workbench (MW) [48], MoTrPAC [38], and 4DN [10, 37] are in the C2M2 format. Although HuBMAP [44] also has a C2M2 package, we instead created a data package from donor, sample, and dataset metadata since more information was available on the data portal.

---

[3]https://datapackage.org/standard/data-package/

[4]https://commonfund.nih.gov/dataecosystem

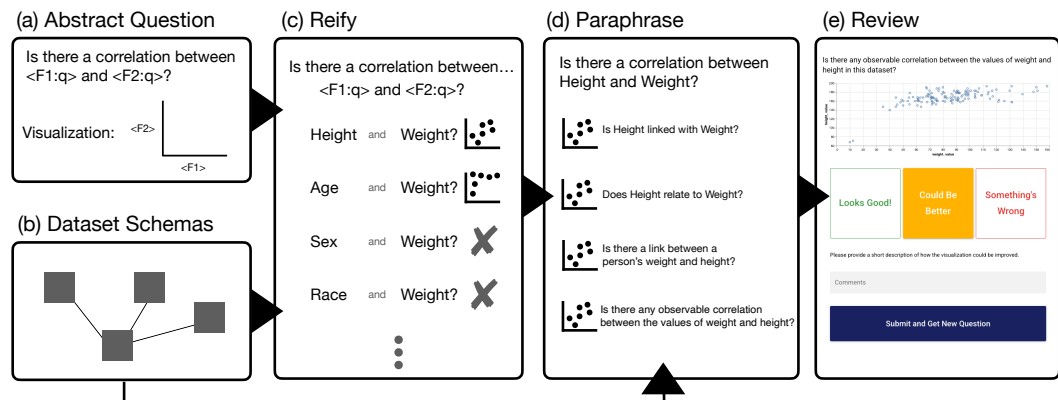

Figure 3: Overview of data synthesis pipeline. (a) Abstract questions are written as templates with placeholders and visualizations that reference those placeholders. (b) Dataset schemas define entity relationship models. (c) Templates are combined with data and placeholders are resolved as long as they satisfy all constraints. (d) The question is paraphrased with an LLM to produce more diverse questions. (e) The final data is reviewed in our review software for issues or potential improvements.

Since HuBMAP includes more data fields than the other data packages, the majority of the queries in DQVis are about HuBMAP (see Table 1).

The natural language $query$ can be in the form of a specific question about the data or an utterance requesting a specific visualization of the data. The query will always include references to entities or fields from the dataset. For instance, "How many donors are there?" would refer to the donor entity, and "What is the average age?" would refer to the age field in the donor entity. A more specified question would be "What is the average age of donors?", but omitting the entity in a query is plausible and can be inferred. It is also possible to ask questions that require multiple entities. For instance, "How many samples are there for each sex?" is asking for the count of records in the sample entity grouped by the sex field in the related donor entity. Queries that are utterances will typically reference a specific chart type, e.g. "Make a scatterplot of height and weight." Such queries can have implied underlying questions, e.g. "Is there a correlation between height and weight?" or can be a step in a more open-ended data exploration process. The answers to these queries for DQVis are visualization specifications ($spec$) in the form of our biomedical visualization grammar.

In addition to the data-question-vis triplets, additional information related to the data creation is recorded in the DQVis dataset. The $query\_template$ and $spec\_template$ values are described in Section 5.1; **query_base** and $constraints$ in Section 5.2; and $expertise$ and $formality$ in Section 5.3.

## 5    Data Synthesis Pipeline

Synthesizing the DQVis dataset of 1.08M data-question-visualization triplets consists four major steps (see Figure 3). First, templates define abstract queries and visualization answers. Next, these are reified with data entities and fields that meet imposed constraints. Then the base query is paraphrased to produce a diverse phrasing of the same question. Finally, the resulting data is reviewed both iteratively to refine and debug issues and to capture expert visualization and domain knowledge.

### 5.1    Abstract Queries and Visualizations

The goal of this pipeline is to capture a wide range of queries that could be posed for a dataset. Our work focuses on low-level analysis tasks that comprise a more involved analysis session. Amar et al. identify ten types of abstract low-level analysis tasks that "cover the vast majority of the corpus of analytic questions [they] studied." [1]. DQVis includes 64 abstract queries that span these ten types of tasks. (Computed Derived Value: 39, Determine Range, 14, Correlate: 4, Cluster: 5, Find Anomalies: 3, Find Extremum: 9, Retrieve Value: 8, Sort: 1, Characterize Distribution: 10, Filter: 10). Some abstract queres relate to multiple low level tasks, especially utterance queries.

These abstract queries and visualizations are in the form of a template with entity and field placeholders. So instead of a dataset specific question "How many biological samples are there for each organ?" the template questions would be "How many <E> are there for each <F>?" where the <E> takes the place of the entity and the <F> takes the place of the data field. This template question is stored as the *query_template* in DQVis. Next, a visualization specification is produced that includes these placeholders, which is stored in the *spec_template* column.

## 5.2 Reify Queries and Visualizations

To reify questions, we replace the entity and field placeholders with concrete entity and field names. However, abstract questions cannot be posed to every possible entity and field. Some questions cannot be logically resolved, like asking what the maximum value is for a nominal field. To account for this, abstract question-vis pairs also include constraints that limit the applicable entities and fields. Constraints are question-specific, though some constraints are applied to multiple questions. There are between 1 and 11 constraints for each question. Every possible question that satisfies all of the constraints is generated with a constraint satisfaction solver library. Depending on the input data and constraints this could lead to an combinatorial explosion of possible solutions. However, for this is not the case for the DQVis dataset and was run on a single personal laptop. The most common type of constraint includes constraints on the field types. Thus, these have a special shorthand in our framework. e.g. "How many <E> are there for each <F:n>?" will add a type constraint that <F> must be a nominal data type. There are additional implied constraints in this question related to entity field relationships. In this example, <F> must exist within <E>. Or, in a more complex case "How many <E1> are there for each <E2.F:n>", <F> exists in <E2> and <E1> and <E2> are two different entities.

Additional constraints are defined as a list in the *constraints* column of DQVis. Each constraint is a boolean expression that allows references to the placeholders (E1, E2, E1.F, E2.F1, E2.F2, etc.) as well as attributes of those entities and fields defined on the data package. For instance, F.c will resolve to the cardinality (unique number of values), in that column. Imposing these constraints helps ensure that the questions are logical and that the visualization responses adhere to visualization design best practices (e.g., limiting the number of distinct categorical values encoded with color). Constraints are also essential for handling different types of data transformations required for the visualization. For instance "How many <E> are there for each <F:n>?' and "How many <E1> are there for each <E2.F:n>" are similar questions with visually similar representations. However, the latter question requires a more complex specification that combines data from multiple data entities. Constraints can ensure that entities have a defined relationship and the type of relationship (e.g., E1 → E2 is one-to-one or many-to-one).

Once constraints are defined, reifying the abstract queries on dataset schemas is formulated as a constraint satisfaction problem. Find all entity-field combinations in the dataset schemas that meet all of the constraints, resulting in a list of solutions that map abstract entities and fields to real entity and field names. Then, for each solution, a data point is created that replaces the template placeholders with these solutions. These are included in the *query_base* column for queries and *spec* for the final specification result.

## 5.3 Query Paraphrasing

Replacing the placeholder entities and fields with real entity and field names will result in real, but repetitive queries, e.g. "How many donors are there for each sex?", "How many donors are there for each race" and "How many samples are there for each organ" will result in some grammatically incorrect queries. E.g. "How many donor are there for each sex". Paraphrasing these queries with an LLM rectifies both of these issues.

We use Ko et al.'s [25] definition for question technicality and formality of language and use a similar technique for paraphrasing across these dimensions We expand the approach to also include the relevant dataset schema in the prompt template for the LLM. Since the dataset schemas include descriptions of entities and fields, this increases the potential of the LLM to generate better paraphrased queries. In addition to varying the style of query, this also replaces exact variable names with synonyms. For instance, if a donor entity has a field named "age_value", we want the phrases "construct a distribution plot of 'age_value'", "show me the age distribution of donors", and "How old are the donors?" to all result in the same plot.

For the DQVis dataset, we used gpt-4o and varied technicality and formality between 1 and 5 exhaustively, resulting in 25 paraphrased $queries$ for each $query\_base$. We also store the $expertise$ and $formality$ scores used to generate these queries.

## 5.4 Multi-Step Generation

To illustrate the potential for DQVis to be applied to multi-step reasoning applications, we provide 11,447 two-step question samples that mimic the realistic user interaction around a data visualization. To implement multi-step question generation with DQVis, we first extracted all single-step queries and their structured $solution$ metadata from our 1.08 million corpus, and deduplicate on the underlying $query\_base$ to isolate prototypical questions. We then designed 17 templates links. When designing the logic to link follow-up questions, we considered models of the information-seeking process summarized in Chapter 3 of Search User Interfaces [17]. One model proposed by Shneiderman et al. 1997 [40] includes four steps. 1. Query Formulation, 2. Action, 3. Review Results, 4. Refinement. The other models have slightly different formulations, but all agree on an interactive cycle where results are acquired and actions are refined based on those results. So in our case, the multi-step questions include an initial question (steps 1 and 2), a resulting visualization (step 3). Then, based on the information presented, a refined follow-up query that requests additional information (Step 4). For example, if the first query asks to view the distribution of donor weight, the second query could ask for the distribution of donor weight grouped by sex. By enforcing matching constraints on entity names, fields, and underlying solution metadata, we collected up to 50 linked question pairs $(Q1, Q2)$ that satisfy the constrains per dataset schema (e.g, HuBMAP, MW, 4DN, MoTrPAC, SenNet), ensuring balanced coverage and yielding 1,273 unique pairs for coherent two-step dialogues. Finally, we apply an LLM paraphrasing step — varying expertise and formality scores in the scale $\{1, 3, 5\}$ — to produce linguistically diverse question pairs. We reduced the number of paraphrased sentences for each multi-step chain and instead prioritized generating more questions. The paraphrasing also introduces diectic phrases for the follow-up query, e.g. group *this* by donor sex. This pipeline can be readily extended to generate multi-round reasoning dialogues of varying complexity, demonstrating DQVis's ability to link broad analytical tasks to detailed follow-ups.

# 6 Data Evaluation

There are multiple ways to evaluate the quality of data-query-visualization triplets. The most basic checks include validation that the visualization specification adheres to our grammar, which is easy to confirm programmatically. However, valid specifications could still result in a malformed or empty visualization or reveal an error in the underlying visualization software. Furthermore, even for well-formed visualizations, there can be better and worse visualizations for a particular question, as well as different individual preferences. Since visualizations are intended for human interpretation, such an evaluation requires a human evaluation. For this evaluation, we recruited five individuals with advanced degrees in computer science, data visualization, biomedical informatics, and professional experience in these domains. All individuals are familiar with HuBMAP and its metadata model. To facilitate this process, we developed an interactive data review interface.

## 6.1 Data Review Interface and Methods

The data review interface allows reviewers to review an individual query and visualization (Figure 3e). For each data point, the reviewer can confirm that this is a reasonable question with a visualization response that satisfies the query. Alternatively, if there is a significant issue, the reviewer can select from a list of predefined options and provide free text feedback on the issue. Finally, when the visualization can still answer the question, but could be improved, the reviewer can select the middle option with a free text suggestion for improvement.

We recruited five individuals with advanced degrees in computer science, data visualization, biomedical informatics, and professional experience in these domains. All individuals are familiar with HuBMAP and its metadata model. In order to compare the similarity of reviewer responses, the first 20 data points are randomly selected once and shown to all reviewers. Then, data points are selected randomly with balance across template queries, resulting in every query template being reviewed. In total, 357 reviews were submitted for 274 unique questions.

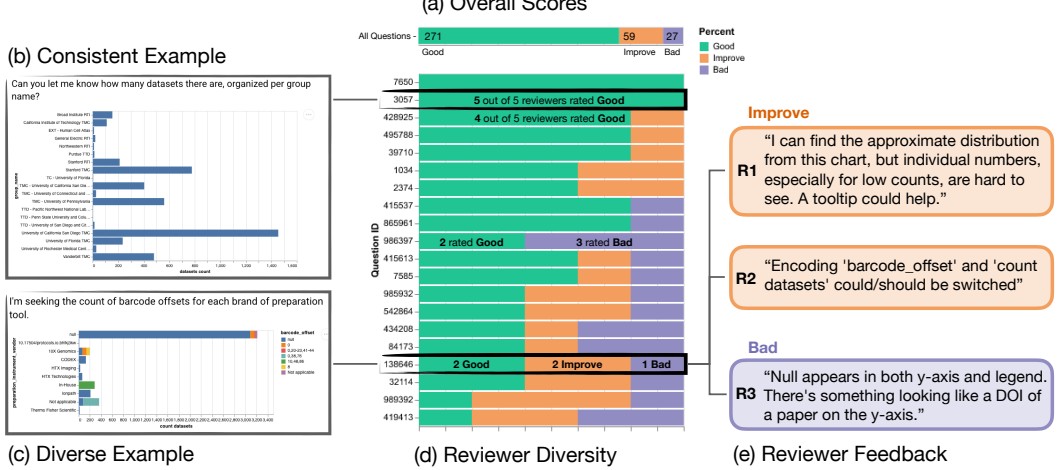

Figure 4: Highlights of review data. (a) Overall scores data was marked positively, with room for improvement. (b) An example of 1 of the 2 questions that was given the same score by all reviewers. (c) An example with 2 positive scores, 2 improve scores, and 1 bad score. (d) The reviewer scores for the 20 overlapping questions all reviewers saw. (e) Qualitative user feedback that points to additional features, different encodings, and data issues — all from the same question.

## 6.2 Results

Overall, 271 out of 357 (75.9%) questions were labeled as good; 59 (16.5%) were marked as needing improvements; and 27 (7.6%) were marked as bad. When viewing the same questions, reviewers responded diversely (See Figure 4d). Of the 20 questions shown to all reviewers, only two had the same score from everyone. The diversity of reviews points to one challenge in designing automated systems for generating visualizations — the diversity of the skills, domain knowledge, visualization knowledge, and preferences of those viewing those visualizations. Reviewers also provided free text descriptions for 73 questions that were marked as 'bad' or 'could be improved'. This feedback provides valuable insight into the process, dataset, and associated tools. The reviewed data is published with the DQVis dataset as an additional resource. Several qualitative themes are present within the reviews. Eleven comments reference the large number of null values, making the visualization difficult to read and implying it would be improved with those values filtered out. One example of this is R3 in Figure 4e, which also calls out a potential issue with the data itself — an unexpected DOI listed as an instrument vendor. Many comments point to issues with the paraphrasing. For instance, the question "Would you please furnish the distribution specifics of the number of input cells or nuclei?", which was paraphrased from "What is the distribution of number_of_input_cells_or_nuclei?" was considered "weird". More importantly, some paraphrased questions changed the meaning of terms. Three comments called out Rh factor as being distinct from blood group or blood type — the variable "rh_blood_group" had been paraphrased to "blood type" or "RH[sic] blood type." In this case, "rh_blood_group" had an empty description in the dataset schema, which may have contributed to this failure. The paraphrasing also changed the requested chart type in some instances, e.g., from "heatmap" to "bar chart" and from "pie chart" to unspecified "chart". Similarly, it altered some aggregation function words, e.g., changing "range of" to "common" values. Other comments point to enhancements for the visualization tool, such as highlighting an entire row in the tabular view or including tooltips (see R1 in Figure 4e). Alternatively, R2 suggests an alternative encodings for the same chart and similar suggestions appear for other charts, such as swapping the x and y axes on a scatterplot. In the scatterplot review, the visualization was still correct, but given the variables, swapping the axes would've conformed more to the convention of placing the independent variable on the x-axis and the dependent variable on the y-axis.

# 7 Discussion

## 7.1 Limitations

Some data fields, or combinations of data fields, result in certain chart types. For instance, placing independent variables on the x-axis or displaying population distributions split by gender as a population pyramid are conventions that visualization designers follow. The template-based approach we take loses these variable-specific conventions. Fortunately, breaking conventions does not make the visualizations inaccurate. Still, DQVis could be further enhanced by including non-template-based data.

The results indicate that the paraphraser can introduce some issues by changing the requested chart types, aggregation functions, and variable names. Hence, the review software included in the dataset generation framework is essential. Although the paraphrasing introduces some imprecision, humans are also imprecise or even incorrect when specifying queries with natural language. Still, more work can be done to characterise imprecise prompts and how best to respond to them.

The DQVis dataset is not balanced with respect to question types, visualization type, or dataset schema. This imbalance is a result of our choice to generate as many questions as possible given our set of constraints and the supplied dataset schemas. In particular, the questions generated from different dataset schemas is impacted by the number of entities and fields in the schema. Although this increases the number of data points and increases the potential of DQVis it also may require users to rebalance the dataset depending on their tasks by subsampling data points from overrepresented categories.

## 7.2 Ethical Considerations

New technology like LLMs introduces the possibility that generated visualizations are incorrect, even ones trained with DQVis. The generation of visualization specifications does provide some inherent guardrails compared to generating images or Python code, since it will always operate within the well-defined bounds of the visualization toolkit. Still, it is possible that data could be transformed incorrectly or presented in a misleading way. Visualization systems that use LLMs to generate visualizations should communicate with users the possibility of such outcomes.

# 8 Conclusion

We introduce DQVis, a dataset of 1.08 million natural language questions and visualization responses for the domain of biomedical research data repositories. This dataset can be used for fine-tuning an LLM for a biomedical natural language interface, potentially enabling critical scientific discoveries. Additionally, it could serve as a reference dataset to benchmark, compare, augment, and synthesize other work in the domain of NL2VIS. Additional domain-specific datasets could utilize the generation and review framework introduced in this work. Such domain-specific datasets have potential for specialized domains that require domain-specific visualizations like body maps and genomics visualizations. Finally, DQVis lays the foundation for multi-step reasoning datasets. By linking together the elemental data points in DQVis, we illustrate how chains of conversation can be constructed.

## Acknowledgments and Disclosure of Funding

This work was supported by ARPA-H AY2AX000028. The authors wish to thank members of the HIDIVE lab for their support and feedback throughout this project, and to the expert reviewers who were essential to this project.

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

# A Technical Appendices and Supplementary Material

## A.1 Code and Resources

The following are the primary DQVis resources.

- Data: `https://huggingface.co/datasets/HIDIVE/DQVis`
- Synthesis Code: `https://github.com/hms-dbmi/DQVis-Generation`
- Review Code: `https://github.com/hms-dbmi/DQVis-review`

A highly-relevant resource that is not contributed by this paper, but used by it is the visualization grammar that DQVis uses. We include the versions used for the creation and review of DQVis.

The **udi-grammar-py** Python package was used for the generation of template specifications.

- Version: 0.2.6
- GitHub: `https://github.com/hms-dbmi/udi-grammar-py`
- PyPI: `https://pypi.org/project/udi-grammar-py/`

The **udi-toolkit** JavaScript package was used in the review software.

- Version: 0.0.24
- GitHub: `https://github.com/hms-dbmi/udi-grammar`
- NPM: `https://www.npmjs.com/package/udi-toolkit`

## A.2 Paraphrasing Prompts

The following is the complete prompt template used for paraphrasing questions.

---

**LLM Prompt Template**

You are a paraphrasing assistant. Your task is to rewrite a given sentence with various styles of language usage. The sentence will either be a question about data, or request to construct a data visualization.

The input sentence will include entity names and fields names from the data. The dataset schema will also be provided to you to enable better paraphrasing of the field and entity names. More technical language may use the exact field names, while more colloquial language may use more general terms, synonyms, and will likely not use the exact field names. e.g. "What is the value of the age_value field?" vs "How old is the person?".
Dataset schema: `{dataset_schema}`

Score-A of 1 indicates a higher tendency to use Colloquial language and a Score-A of 5 indicates a higher tendency to use Standard language.
Score-B of 1 indicates a higher tendency to use Non-technical language and a Score-B of 5 indicates a higher tendency to use Technical language.
Rewrite the following sentence as if it were spoken by a person with a given score for language usage.

Sentence: `{sentence}`

##
Score-A 1, Score-B 1: Score-A 1, Score-B 2: Score-A 1, Score-B 3: Score-A 1, Score-B 4: Score-A 1, Score-B 5: Score-A 2, Score-B 1: Score-A 2, Score-B 2: Score-A 2, Score-B 3: Score-A 2, Score-B 4: Score-A 2, Score-B 5: Score-A 3, Score-B 1: Score-A 3, Score-B 2: Score-A 3, Score-B 3: Score-A 3, Score-B 4: Score-A 3, Score-B 5: Score-A 4, Score-B 1: Score-A 4, Score-B 2: Score-A 4, Score-B 3: Score-A 4, Score-B 4: Score-A 4, Score-B 5:

---

Score-A 5, Score-B 1: Score-A 5, Score-B 2: Score-A 5, Score-B 3: Score-A 5, Score-B 4: Score-A 5, Score-B 5:

| Placeholder | Example Value / Description |
|---|---|
| {dataset_schema} | A JSON schema showing dataset fields and entity types. |
| {sentence} | The input text to paraphrase, e.g., `"Is there a correlation between age_value and weight_value?"` |

Table 2: Placeholders and example input values used in the LLM paraphrasing prompt template.

In addition to the prompt we use structured outputs to require a list of paraphrased sentences with a formality and expertise score. The description of the formality and expertise score are included in the structured output definition.

**Formality.** Colloquial (Score=1) language is informal and used in everyday conversation, while standard language (Score=5) follows established rules and conventions and is used in more formal situations.

**Expertise.** Technical language (Score=5) is often used by experts in a particular field and includes specialized terminology and jargon. Non-technical language (Score=1), on the other hand, is more accessible to a general audience and avoids the use of complex terms.

### A.3 Reviews

This section shows all of the data that is summarized in Figure 4. These are not all the results collected, but just the 20 data points that every reviewer saw. All review data is available on the Hugging Face repository. Each page in this section includes the question ID, the query, and the visualization shown to the reviewers, along with all five reviewer responses. The review interface allowed users to select a score of good, improve, or bad. If the selection was not good, then the user could select predefined issue categories and leave free-text comments.

**Question ID: 7650**

**Query:** Can you tell me the number of subject categories we've got, broken down by the level of detail?

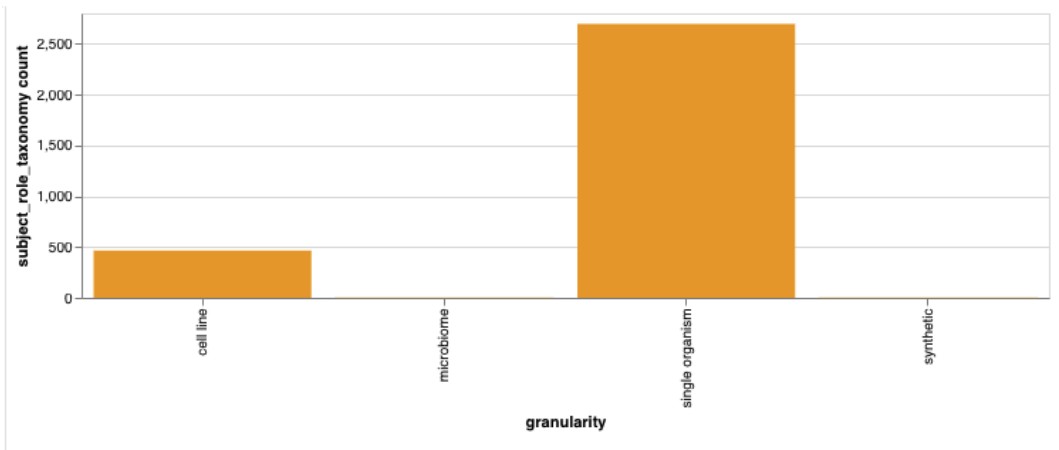

| Reviewer | Score | Issue Categories | Comments |
|----------|-------|------------------|----------|
| R1 | Good | | |
| R2 | Good | | |
| R3 | Good | | |
| R4 | Good | | |
| R5 | Good | | |

**Question ID: 3057**

**Query:** Can you let me know how many datasets there are, organized per group name?

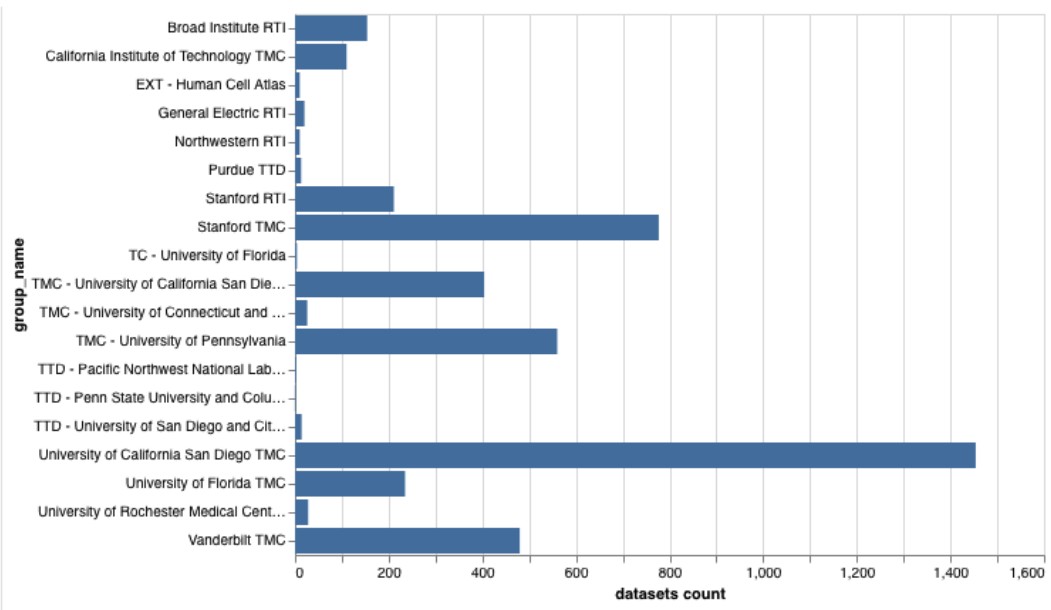

| Reviewer | Score | Issue Categories | Comments |
|----------|-------|------------------|----------|
| R1 | Good | | |
| R2 | Good | | |
| R3 | Good | | |
| R4 | Good | | |
| R5 | Good | | |

**Question ID: 428925**

**Query:** Could you highlight the record with the lowest RNA-seq input amount listed?

| rnaseq_assay_input | hubmap_id | uuid | ablation_distance_bet... | ablation_distance_bet... |
|---|---|---|---|---|
| 356 | HBM477.KVFD.827 | e79b021fd60b54850cbb5bf | ∅ | ∅ |
| 407 | HBM354.GTPP.329 | 1d154d589fb8cffbb8d1f056 | ∅ | ∅ |
| 523 | HBM644.LHFR.583 | 1a7c7284d18dd06273dba58 | ∅ | ∅ |
| 666 | HBM439.LWSZ.467 | 4f20058fbdca73ddbf8a0fe4 | ∅ | ∅ |
| 1488 | HBM574.NFCS.842 | ec88a6b161dce97a2361b14 | ∅ | ∅ |
| 2045 | HBM453.GWNF.247 | b34aa1ec24b8447eee71053 | ∅ | ∅ |
| 2102 | HBM487.WJST.938 | 0c36dd5c4e727cfd2efd806 | ∅ | ∅ |
| 2178 | HBM379.PCLL.836 | 27b0957d7c43322c272882 | ∅ | ∅ |
| 2296 | HBM949.PNXL.623 | 025e083e54722e695cdecd | ∅ | ∅ |
| 2516 | HBM854.LQKL.226 | 9b048a63ac274e36942d49 | ∅ | ∅ |
| 2865 | HBM727.CLDW.546 | b3a0cf5d7e85cc77f50d1bfd | ∅ | ∅ |
| 2999 | HBM322.TNGF.859 | 2e6c312200bea94f832c96 | ∅ | ∅ |
| 3139 | HBM367.ZMBH.758 | d611a7de3a07bd5b88e669 | ∅ | ∅ |
| 3358 | HBM958.VZLG.297 | 986c769e5fe01c550b75e4 | ∅ | ∅ |
| 3730 | HBM233.XQZM.395 | 63325f48a2b8ab0564617a | ∅ | ∅ |
| 3920 | HBM375.ZKZZ.765 | cfc125d6d916f121e92a840 | ∅ | ∅ |
| 4000 | HBM475.NWHG.922 | b4a975cb708bf442ceeb4ad | ∅ | ∅ |
| 4000 | HBM846.NMQR.693 | ee14de43eba29d0c55481a | ∅ | ∅ |
| 4000 | HBM398.ZSNW.578 | d74c1643f3f4c22a4758e59 | ∅ | ∅ |
| 4020 | HBM684.SLGB.599 | 64e3949e4a4cc433e64745 | ∅ | ∅ |
| 4080 | HBM793.LCCQ.642 | 4c26f91beabafb3290fad2bf | ∅ | ∅ |
| 4113 | HBM925.FODP.328 | 1f9e84f5306c1cc75db60184 | ∅ | ∅ |

| Reviewer | Score | Issue Categories | Comments |
|---|---|---|---|
| R1 | Good | | |
| R2 | Good | | |
| R3 | Good | | |
| R4 | Good | | |
| R5 | Improve | | maybe highlight the whole row, but this will do. |

**Question ID: 495788**

**Query:** What's the complete spread of library concentration values for each assay category?

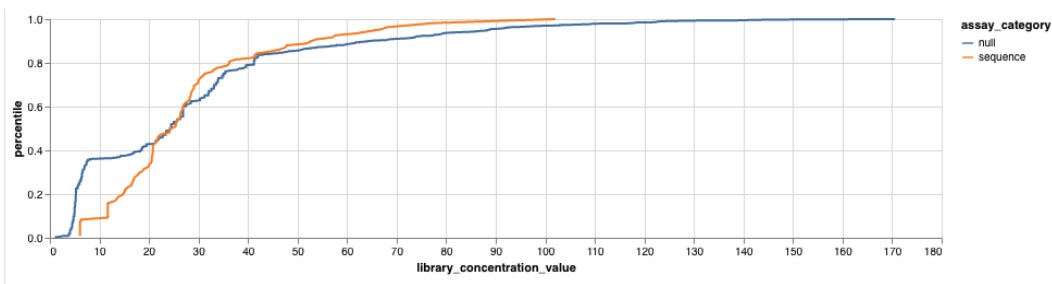

| Reviewer | Score | Issue Categories | Comments |
|---|---|---|---|
| R1 | Improve | | library_concentration_value is missing unit |
| R2 | Good | | |
| R3 | Good | | |
| R4 | Good | | |
| R5 | Good | | |

**Question ID: 39710**

**Query:** Could you give me the number of datasets grouped by the fields 'umi_read' and 'sc_isolation_protocols_io_doi'?

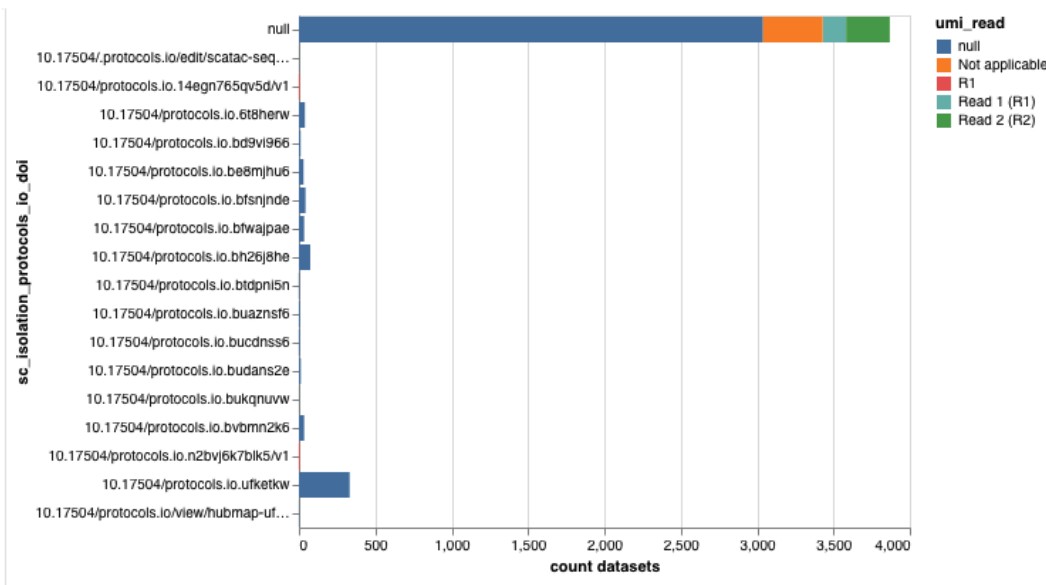

| Reviewer | Score | Issue Categories | Comments |
|---|---|---|---|
| R1 | Good | | |
| R2 | Improve | | null value again messes readability of the visualization |
| R3 | Good | | |
| R4 | Good | | |
| R5 | Good | | |

**Question ID: 1034**

**Query:** Can you identify how many datasets are grouped by the respective units used for library concentration?

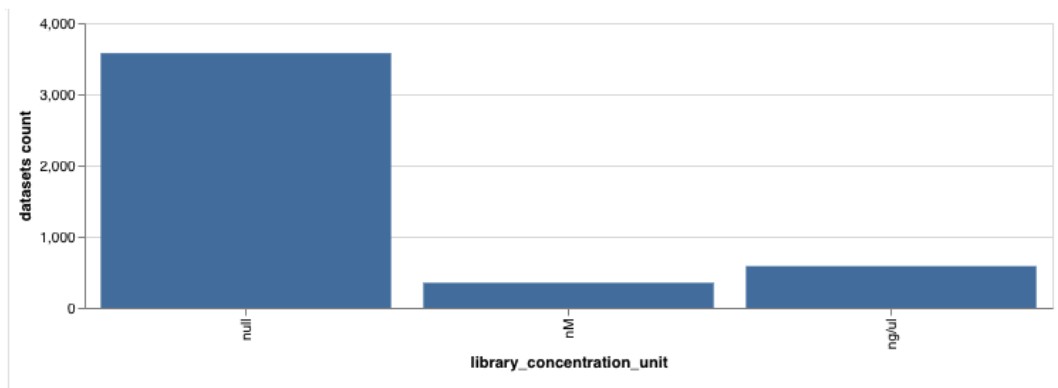

| Reviewer | Score | Issue Categories | Comments |
|----------|---------|------------------|----------|
| R1 | Good | | |
| R2 | Improve | | Approximately: yes, but finer grid would be easier to get more precise values |
| R3 | Good | | |
| R4 | Improve | | Can estimate but with only bars and grid lines every 1000 units will be rough |
| R5 | Good | | |

**Question ID: 2374**

**Query:** Could you determine the number of datasets, segmented based on sc_isolation_protocols_io_doi identifiers?

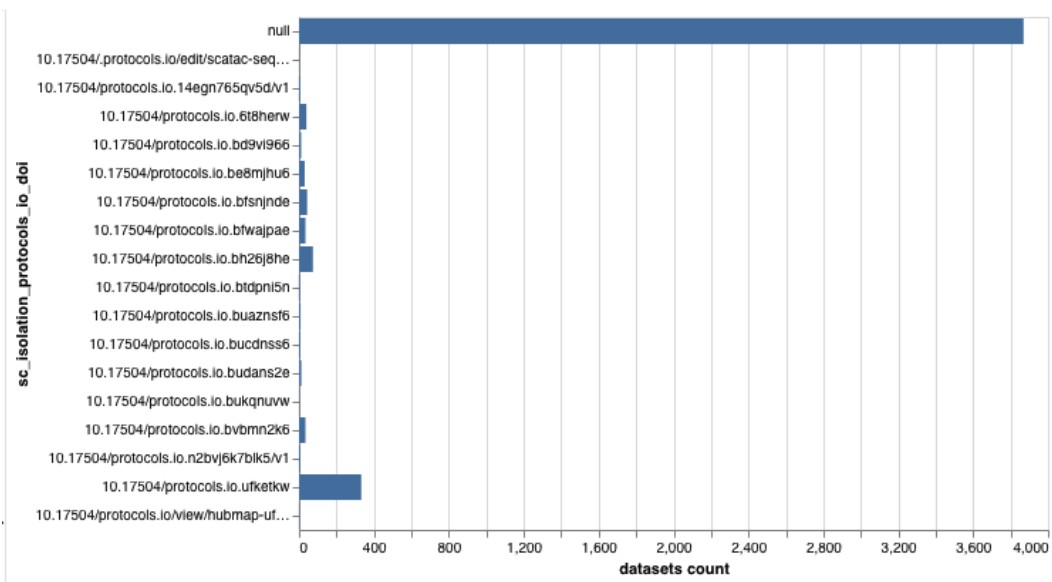

| Reviewer | Score | Issue Categories | Comments |
|---|---|---|---|
| R1 | Good | | |
| R2 | Improve | | null value leads to very difficult distinction of the other (actual) values, impossible to get a precise number |
| R3 | Improve | | Not sure what segmented means in this context |
| R4 | Good | | |
| R5 | Good | | |

**Question ID: 415537**

**Query:** Could you kindly create a circular diagram highlighting the transposition transposase origin?

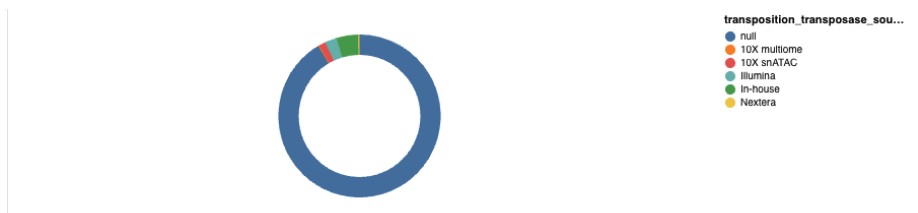

| Reviewer | Score | Issue Categories | Comments |
| --- | --- | --- | --- |
| R1 | Good | | |
| R2 | Bad | Other | null value breaks the visualization + not sure whether "circular diagram" = donut chart |
| R3 | Good | | |
| R4 | Good | | |
| R5 | Good | | |

**Question ID: 865961**

**Query:** Are there clusters of storage duration and library output values based on different chromatin capture methods?

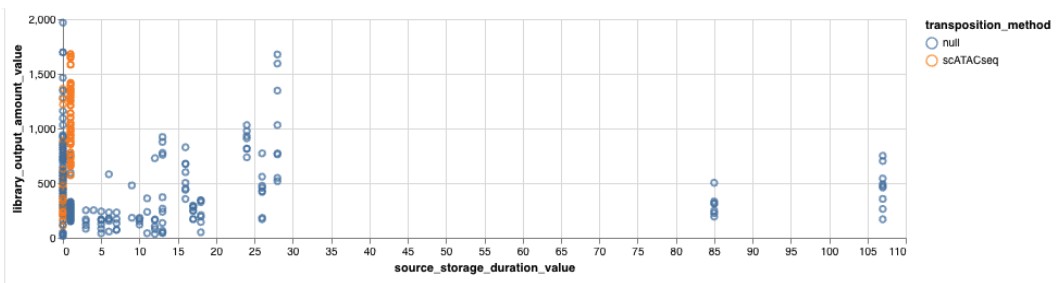

| Reviewer | Score | Issue Categories | Comments |
|---|---|---|---|
| R1 | Bad | Other | transposition_method is chromatin capture methods? given that there only is 1 (and null), can't really answer this
also again time value is missing unit |
| R2 | Good | | |
| R3 | Good | | |
| R4 | Good | | |
| R5 | Good | | |

**Question ID: 986397**

**Query:** Could you describe the pattern in step_z_value distribution?

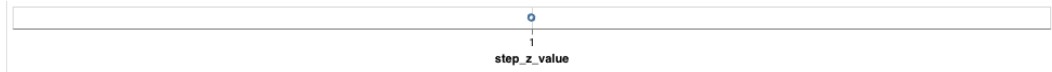

| Reviewer | Score | Issue Categories | Comments |
|---|---|---|---|
| R1 | Bad | Malformed Visualization, Question Not Answered | |
| R2 | Bad | Malformed Visualization | |
| R3 | Bad | Malformed Visualization | Only one data point visible |
| R4 | Good | | |
| R5 | Good | | |

**Question ID: 415613**

**Query:** Could you assemble a donut chart detailing the tile arrangement?

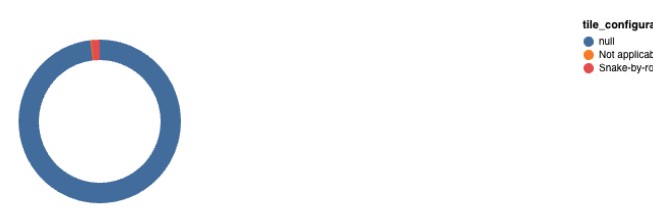

| Reviewer | Score | Issue Categories | Comments |
|---|---|---|---|
| R1 | Good | | |
| R2 | Bad | Other | null value overtakes whole visualization |
| R3 | Improve | | Need percentage number for detailing |
| R4 | Good | | |
| R5 | Good | | |

**Question ID: 7585**

**Query:** How many dataset bundles are there when arranged by blood type?

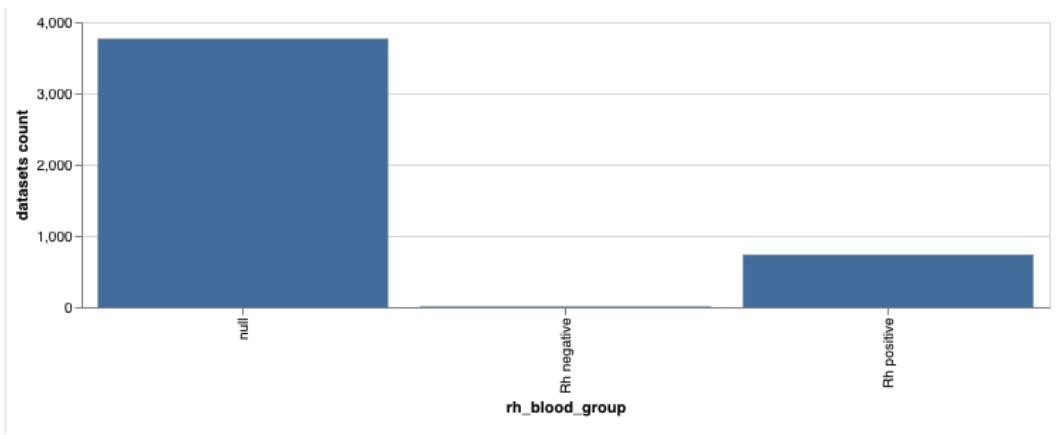

| Reviewer | Score | Issue Categories | Comments |
| --- | --- | --- | --- |
| R1 | Bad | Question Not Answered | Visualization just showing rhesus factor, not blood group |
| R2 | Good | | |
| R3 | Good | | |
| R4 | Improve | | Rh factor is not the same as blood type |
| R5 | Good | | |

**Question ID: 985932**

**Query:** Would you please furnish the distribution specifics of the number of input cells or nuclei?

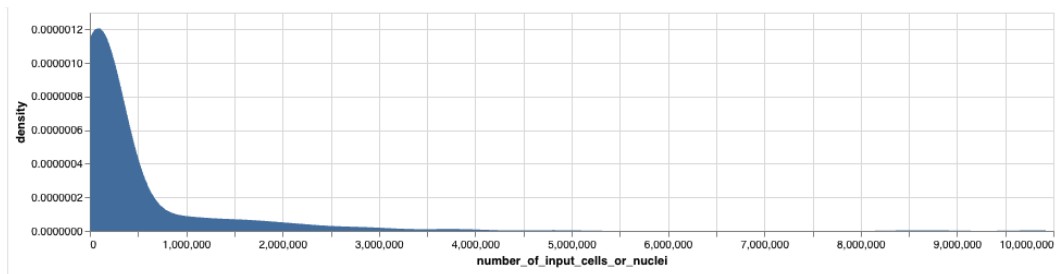

| Reviewer | Score | Issue Categories | Comments |
|---|---|---|---|
| R1 | Improve | | decent visualization, but how can the number of input cells/nuclei be negative? |
| R2 | Bad | Bad Question | |
| R3 | Improve | | question wording sounds weird |
| R4 | Good | | |
| R5 | Good | | |

**Note**: This figure is slightly different from the version shown to the participant. In an earlier version of the visualization toolkit the visualization was not clipped at zero. The visualization itself has not changed.

**Question ID: 542864**

**Query:** Are there any visible clusters in datasets concerning library_adapter_sequence and time items were preserved?

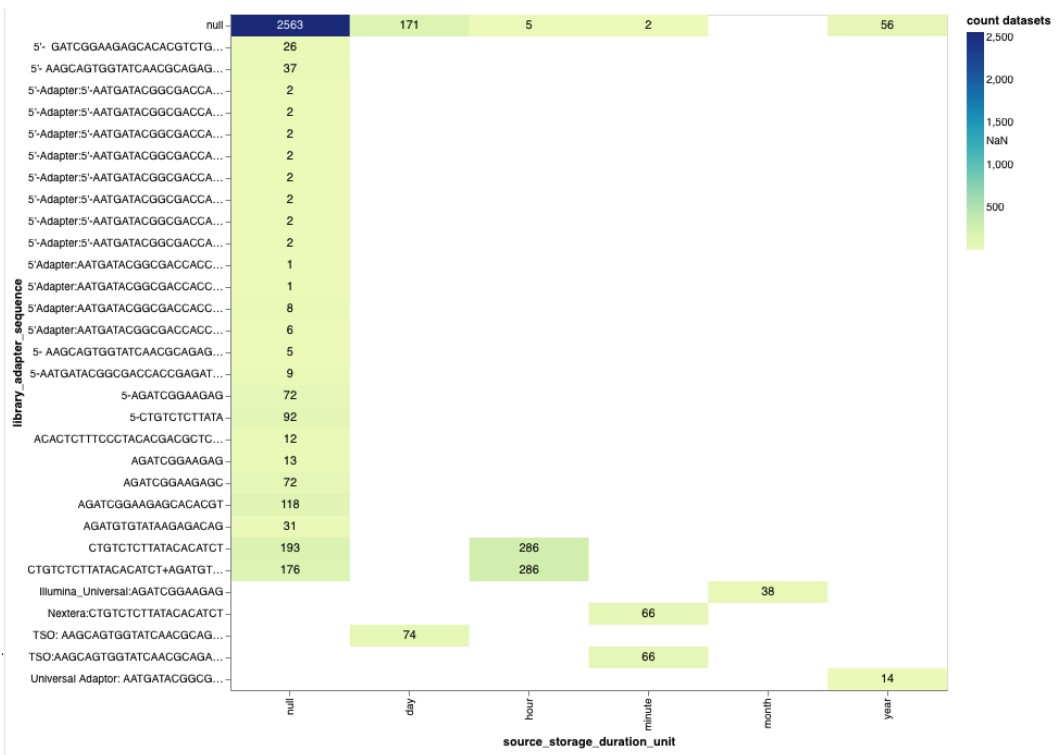

| Reviewer | Score | Issue Categories | Comments |
|---|---|---|---|
| R1 | Bad | Malformed Visualization | time is grouped by unit, rather than time amount |
| R2 | Good | | |
| R3 | Improve | | The value for null badly skewed the color range |
| R4 | Good | | |
| R5 | Improve | | It was difficult for me to connect the two, i.e., color was showing clusters. Maybe colors could be more differentiable. or use shape as a visual encoding as well. |

**Question ID: 434208**

**Query:** Can you sort these datasets by the gap in Z coordinates?

| increment_z_value | hubmap_id | uuid | ablation_distance_bet... | ablation_distance_bet... | |
|---|---|---|---|---|---|
| true | HBM423.JZTB.864 | 073cad035ce246a0134e22′ | ∅ | ∅ | |
| true | HBM675.SDNC.963 | 298caad597d4a9eaaa3edb( | ∅ | ∅ | |
| true | HBM384.XMBW.725 | b6eba6afe660a8a85c2648e | ∅ | ∅ | |

| Reviewer | Score | Issue Categories | Comments |
|---|---|---|---|
| R1 | Bad | Question Not Answered | |
| R2 | Bad | Other | the z coordinate of interest is either boolean, or somewhere we I need to scroll too far to the right, which makes it difficult to validate the question at first glance. |
| R3 | Good | | |
| R4 | Improve | | Columns could be sorted to reduce horizontal scrolling, as there are many columns and I am not sure which corresponds to "gap in Z coordinates" |
| R5 | Good | | |

**Question ID: 84173**

**Query:** Identify the distinct frequency for barcode reads within varying unique molecular index sizes.

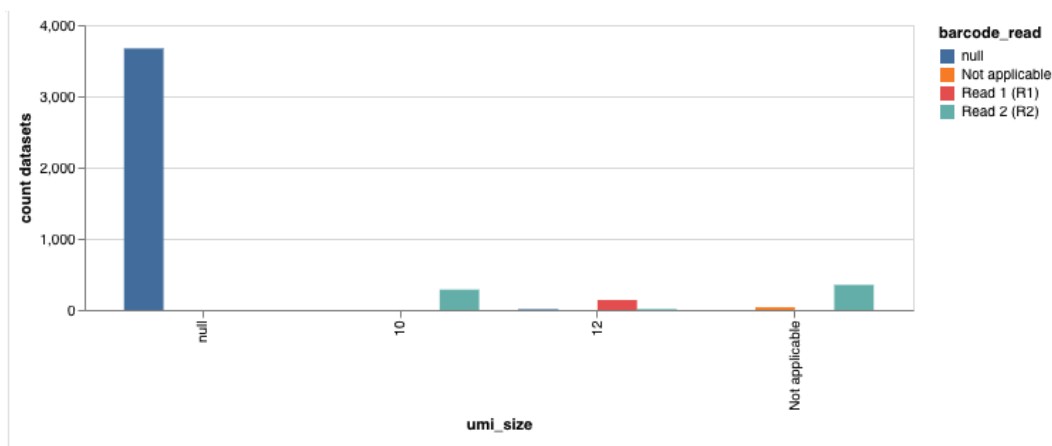

| Reviewer | Score | Issue Categories | Comments |
|---|---|---|---|
| R1 | Bad | | |
| R2 | Improve | | Low count items almost indistinguishable, could use different y-axis scale. |
| R3 | Bad | Malformed Visualization, Other | The bars are not aligned with ticks. There are also two bars with the same colors. |
| R4 | Good | | |
| R5 | Good | | |

**Question ID: 138646**

**Query:** I'm seeking the count of barcode offsets for each brand of preparation tool.

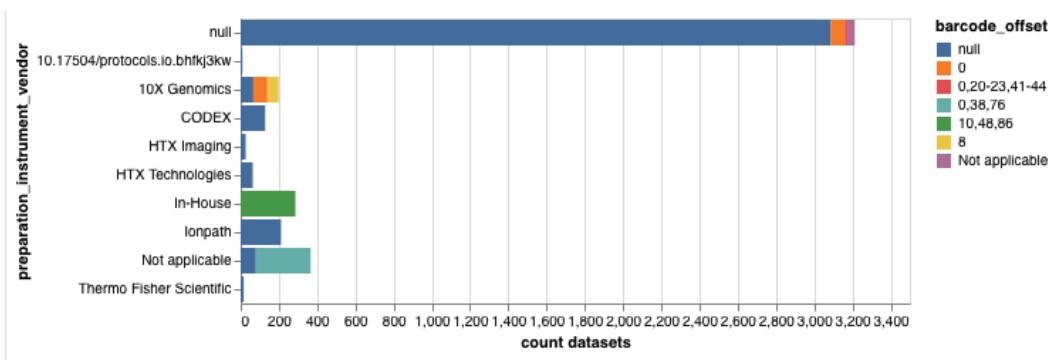

| Reviewer | Score | Issue Categories | Comments |
|---|---|---|---|
| R1 | Improve | | I can find the approximate distribution from this chart, but individual numbers, especially for low counts, are hard to see. A tooltip could help. |
| R2 | Improve | | Encoding 'barcode_offset' and 'count datasets' could/should be switched |
| R3 | Bad | Other | Null appears in both y-axis and legend. There's something looking like a DOI of a paper on the y-axis. |
| R4 | Good | | |
| R5 | Good | | |

**Question ID: 32114**

**Query:** Please provide the dataset count, categorized by consortium and kind of signal.

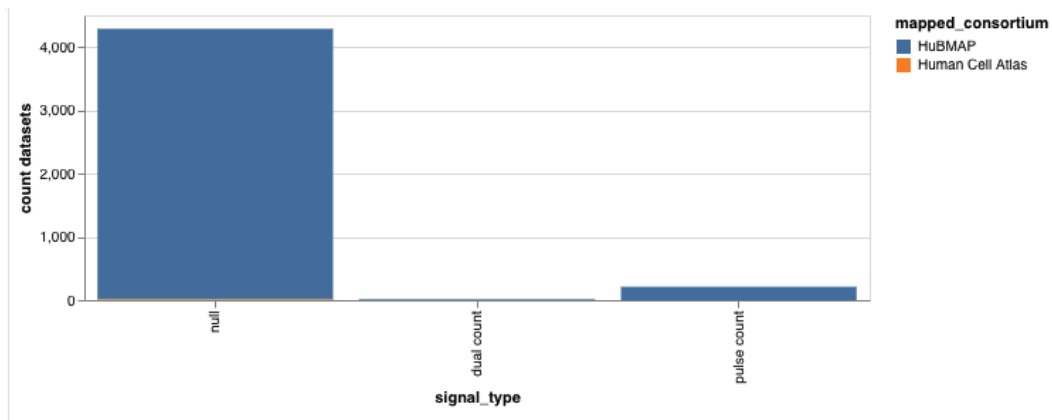

| Reviewer | Score | Issue Categories | Comments |
| --- | --- | --- | --- |
| R1 | Bad | Question Not Answered | HCA is mentioned in the legend but not shown in the visualization. |
| R2 | Improve | | the hubmap cell atlas data are almost not visible |
| R3 | Improve | | Need specific number annotated to each bar to be able to provide specific dataset count |
| R4 | Good | | |
| R5 | Good | | |

**Question ID: 989392**

**Query:** Is the storage duration consistent with each barcode length?

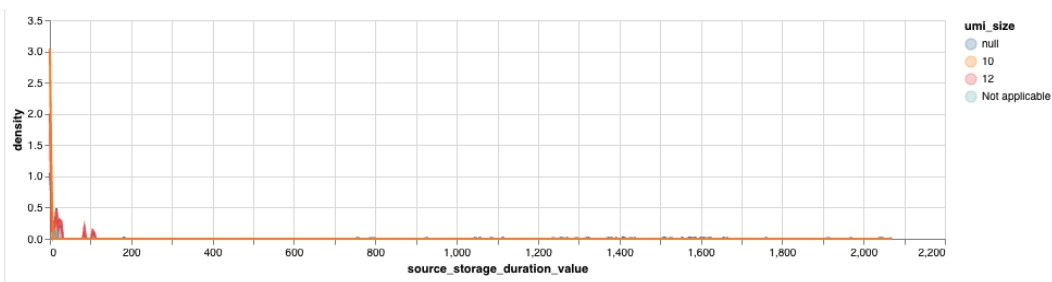

| Reviewer | Score | Issue Categories | Comments |
|----------|---------|----------------------|----------|
| R1 | Good | | |
| R2 | Improve | | Difficult to answer |
| R3 | Bad | Question Not Answered | |
| R4 | Improve | | What does it mean by "consistent"? Also, I cannot distinguish the different lines from each other visually. |
| R5 | Improve | | I see distribution, rather than answer for consistency. If correlation is meant, maybe a different graph would be helpful? |

**Question ID: 419413**

**Query:** How does the data for taxonomy numbers like those found in NCBI appear?

| id | clade | name | description |
|---|---|---|---|
| NCBI:txid10009 | species | Tamiasciurus hudsonicus | red squirrel |
| NCBI:txid10029 | species | Cricetulus griseus | Chinese hamsters |
| NCBI:txid10036 | species | Mesocricetus auratus | Syrian hamster |
| NCBI:txid1006131 | species | Tetrastigma loheri | ∅ |
| NCBI:txid10090 | species | Mus musculus | mouse |
| NCBI:txid10116 | species | Rattus norvegicus | rats |
| NCBI:txid10149 | species | Hydrochoerus hydrochaeris | carpincho |
| NCBI:txid1093657 | species | Pitcairnia flammea | ∅ |
| NCBI:txid110662 | species | Synechococcus sp. CC9605 | ∅ |
| NCBI:txid112262 | subspecies | Ovis canadensis canadensis | ∅ |
| NCBI:txid112509 | subspecies | Hordeum vulgare subsp. vulg | two-rowed barley |
| NCBI:txid1129 | genus | Synechococcus | ∅ |
| NCBI:txid1148 | species | Synechocystis sp. PCC 6803 | ∅ |
| NCBI:txid1280 | species | Staphylococcus aureus | ∅ |
| NCBI:txid1282 | species | Staphylococcus epidermidis | ∅ |
| NCBI:txid129788 | species | Ruditapes philippinarum | Japanese littleneck |
| NCBI:txid1309 | species | Streptococcus mutans | ∅ |
| NCBI:txid1314 | species | Streptococcus pyogenes | ∅ |
| NCBI:txid132113 | species | Bombus impatiens | ∅ |
| NCBI:txid1351 | species | Enterococcus faecalis | ∅ |
| NCBI:txid13821 | species | Pteris vittata | ∅ |
| NCBI:txid1408 | species | Bacillus pumilus | ∅ |

| Reviewer | Score | Issue Categories | Comments |
|---|---|---|---|
| R1 | Improve | | Question is vague - "how does the data appear" - so unsure if the table answers this question. |
| R2 | Bad | Bad Question | |
| R3 | Bad | Bad Question | |
| R4 | Good | | |
| R5 | Improve | | Maybe this should be phrased like, give me a sample or summary of the data. When I see the question, I'd expect this to give me some analysis rather than a dataset? |

