# OpenReview forum: "DQVis Dataset: Natural Language to Biomedical Visualization"
_NeurIPS.cc/2025/Datasets_and_Benchmarks_Track — NeurIPS 2025 Datasets and Benchmarks Track poster_

### Official Review · Reviewer_wTno · 2025-06-01

**Rating:** 4
**Confidence:** 4

**Summary:**

DQVis is a large-scale, domain-specific NL‑to‑visualization benchmark for biomedical research data portals. Using a template‑and‑constraint framework, the authors generate 1.08 million data‑question‑visualization triplets drawn from five public biomedical repositories (HuBMAP, Metabolomics Workbench, 4DN, MoTrPAC, SenNet) and 11 447 two‑step query pairs to support multi‑step reasoning. Each triplet consists of (1) a structured data schema, (2) a natural language question about that schema, and (3) a JSON visualization spec in a custom biomedical grammar. The paper also includes an open‑source review interface, through which five domain experts rated a random sample of 357 visualizations; 75.9 % were marked “good,” 16.5 % “needs improvement,” and 7.6 % “bad.” Generation code, review code, and the dataset itself are freely available on Hugging Face and GitHub.

**Dataset Code Accessibility:**

Yes

**Ethical Considerations:**

No, there are no or only very minor ethics concerns

**Final Justification:**

accept

**Limitations Weaknesses:**

Template‑Driven Semantics May Omit Conventions: The template‑based approach sacrifices certain visualization conventions (e.g., population pyramids for age/gender splits, axis orientation norms). Consequently, some generated specs violate best practices—though they remain functionally “correct.” Section 7.1 acknowledges that non‑template data could mitigate this.

Paraphrasing Imperfections: Expert reviews identified instances where GPT‑4o paraphrases changed chart types (e.g., “heatmap” → “bar chart”) or misrepresented field semantics (e.g., “rh_blood_group” paraphrased as “blood type”), leading to mismatches between question intent and visualization. While essential to increase linguistic variety, such errors suggest the need for more refined prompts or schema descriptions.

Data‑Sampling Bias and Representativeness: DQVis draws heavily on HuBMAP metadata (Table 1 shows HuBMAP accounts for the majority of chart complexity examples). The paper does not quantify the geographic, demographic, or technology biases inherent in each repository’s metadata. As a result, models trained on DQVis may overfit to HuBMAP‑style schemas and underperform on less‑represented portals.

**Strengths Contributions:**

Multi‑Step Reasoning Samples: By constructing 11 447 two‑step question pairs—linking broad prompts to more detailed follow‑ups—the dataset explicitly supports conversational and iterative data exploration. Such multi‑step chains are rare in existing resources.

---

> ### Author Rebuttal · Authors · 2025-07-30
>
> Thank you for your review of DQVis, your kind words regarding the multi-step reasoning samples, and your identification of limitations. Please feel free to reach out with additional questions, comments, or feedback; we appreciate your suggestions for improving the paper.
>
> **Summary:** The review restates two limitations discussed in the limitations section and includes a third weakness on the imbalanced nature of the data. We elaborate here on why the contributions of DQVis still warrant publication and how we will update the paper to improve the communication of the strengths and weaknesses of our approach.
>
> **W1 & W2: Template-Based Limitations & Paraphrasing Imperfections**
> The review recapitulates the limitations of template-based approaches and paraphrasing that we describe in Section 7.1 Limitations.
>
> Existing work in the domain illustrates the value of template-based approaches. For instance, an existing dataset paper for general natural language to visualization — *“Synthesizing Natural Language to Visualization (NL2VIS) Benchmarks from NL2SQL Benchmarks”,* Y. Luo, N. Tang, G. Li, C. Chai, W. Li, and X. Qin, in *Proceedings of the 2021 International Conference on Management of Data*, in SIGMOD ’21. doi: 10.1145/3448016.3457261 *— (NVBench),* transforms a natural language to a SQL dataset, which uses a template-based approach for generating data. Our approach goes beyond this existing work by applying modern LLMs for question paraphrasing.
>
> Paraphrasing with LLMs is currently the most effective and scalable method available, despite some known limitations. We adapt the approach proposed in — “Natural Language Dataset Generation Framework for Visualizations Powered by Large Language Models”, H.-K. Ko et al., in Proceedings of the 2024 CHI Conference on Human Factors in Computing Systems, in CHI ’24, doi: 10.1145/3613904.3642943 — which utilizes LLMs for paraphrasing natural language queries.
>
> We extend the approach in two key ways:
>
> 1. For paraphrasing of multi-step questions, our implementation provides detailed guidelines for the LLM, explicitly instructing it to preserve exact field names at higher formality/expertise levels and avoid unnecessary synonymization.
>
> 2. We also pass structured schema information alongside each paraphrasing task, enabling the LLM to refer to the dataset’s canonical field names and structure. This context supports accurate interpretation and reduces the risk of field name misrepresentation.
>
> The data we collected from five reviewers indicates that, the paraphraser’s mistakes do not dominate the dataset. Out of 357 reviewed examples, only 27 (7.6 %) were rated as bad (Figure 4b in our paper). Many of the problematic paraphrases stemmed from missing field descriptions (e.g. “rh_blood_group” lacked a definition in the metadata), which made it difficult for GPT‑4 to choose a meaningful synonym.
>
> It is worth noting that many natural language interfaces must handle user queries that are imprecise or even incorrect. Our dataset intentionally includes linguistic variety to help models handle such scenarios.
>
> In short, we argue that while templating and paraphrasing have limitations, they represent the current state-of-the-art approach, and we have take steps to mitigate the limitations and evaluate the results.
>
>
> **W3: Unbalanced Data Generation**
> DQVis includes data from five biomedical data repositories. The number of questions generated for each of these repositories is reported in Table 1, and the review points out that the majority of questions come from one of the five repositories (HuBMAP).
>
> We believe balancing the dataset during generation is not necessary and would be counterproductive since it would limit the applicability of the dataset. Instead, we recommend that users subsample or rebalance the dataset as appropriate for their task (e.g., by visualization type or dataset schema).
>
> Our approach is to generate as many questions as possible given our set of constraints and the supplied dataset schemas. This approach will generate an imbalance in the dataset due to differences in the dataset schemas, especially the number of entities and fields in the schema. We could subsample the space of possible valid questions to achieve more balance; however, this would reduce the number of generated data points.
>
> We will add a paragraph to the limitations section clarifying that the dataset is not balanced by design and recommending that users perform any desired rebalancing during model training or evaluation. We will provide guidance on how to accomplish this based on the provided templates and constraints.
>
> Finally, we emphasize that while this manuscript introduces the DQVis dataset, we are also proposing a novel dataset generation framework. This framework can easily incorporate additional dataset schemas. In the future, as additional, more diverse biomedical dataset schemas become available, they could be incorporated into the DQVis dataset or used to generate a new one.

---

### Official Review · Reviewer_Zn66 · 2025-06-30

**Rating:** 4
**Confidence:** 3

**Summary:**

This paper presents a framework for automatically generating natural language interfaces to biomedical data repositories through interactive visualizations. The framework maps abstract, low-level data queries and corresponding visualization grammar specifications to concrete instances by selecting suitable data entities and fields, paraphrasing the queries, and producing data-question-visualization triplets. Then the work uses the pipeline to construct the benchmark.

**Dataset Code Accessibility:**

Yes

**Ethical Considerations:**

No, there are no or only very minor ethics concerns

**Final Justification:**

The author has addressed some of my concerns. But for a benchmark paper, all details should be clear.

**Limitations Weaknesses:**

1. The details of the data synthesis process are unclear. Specifically, it is not clear how many Abstract Queries are used in the experiments or whether these queries are sufficient to cover the full range of possible query types.

2. The contribution of the proposed benchmark is not clearly. Although the paper claims to focus on a specific domain, the task—mapping natural language to charts—has already been extensively explored in prior work. Therefore, it would be beneficial to conduct additional experiments to better highlight the unique challenges posed by the constructed benchmark.

3. The performance of VLMs on the proposed benchmark is not clearly reported.

4. The details of the human evaluation process are unclear. In particular, inter-annotator agreement among the five annotators is not reported, making it difficult to assess the reliability of the annotations.

**Strengths Contributions:**

1. The paper propose a data synthesis method and builds the benchmark DQVis.

2. The benchmark contains 1.08 million triplets and 11.4K two-step query samples.

3. The paper conduct an open-source review interface was developed for expert evaluation. Then the work employ five visualization experts to give feedbacks for data quality estimation.

---

> ### Author Rebuttal · Authors · 2025-07-30
>
> Thank you for your time reviewing DQVis. We appreciate your positive comments on the dataset, review interface, and evaluation, as well as your suggestions for improvement. Your review will help us craft the best possible version of this paper.
>
> **Summary:** The review asks for clarification on some details from the paper. We provide answers here and will update the paper to improve the clarity of our communication.
>
> **W1 Abstract Queries Counts**
> The dataset contains 64 template questions. We will include this information either in text or as a table summarizing the counts across different analytical task types. The table will either be included in the paper or supplemental material, depending on space constraints.
>
> **W2 Paper Contribution Clarification**
> There are two main contributions of this paper: a data-question-visualization generation framework and a new dataset (DQVis) of biomedical research repositories constructed with this framework.
>
> The review asks for clarification of the dataset contribution with respect to existing work mapping natural language to visualizations. Although there is overlap with existing work, we believe there is enough of a distinction in DQVis with respect to scale, domain, and inclusion of multi-step queries to warrant publication.
>
> An important work to compare against is an existing dataset of natural language queries and visualizations, *“Synthesizing Natural Language to Visualization (NL2VIS) Benchmarks from NL2SQL Benchmarks”,* Y. Luo, N. Tang, G. Li, C. Chai, W. Li, and X. Qin, in *Proceedings of the 2021 International Conference on Management of Data*, in SIGMOD ’21. doi: 10.1145/3448016.3457261 — (**NVBench**).
>
> *Scale*: NVBench includes 25,750 datapoints, whereas DQVis includes 1,075,190 datapoints, a more than 40x increase in scale.
>
> *Multi-Step*: DQVis contains multi-step data, which is increasingly vital for conversational LLMs. NVBench does not include multi-step data.
>
> *Domain*: DQVis is a domain-specific dataset centered around biomedical research repositories, whereas NVBench is a general-purpose dataset. While there is overlap and advantages to both approaches, there is value in domain-specific datasets. One advantage is that the reviewers are familiar with both visualization design and the domain of the data, and are thus able to review the quality of this data in a way that a general-purpose audience cannot.
>
> These distinctions support the value of DQVis as a dataset for future research. We will add these comparisons to the related work section.
>
> **W3 VLM Performance**
> The review asks for the performance of VLMs to be reported in the paper; however, we do not train any VLMs in DQVis. To clarify, there are two main contributions of this paper:
>
> 1. A framework for generating datasets of natural language queries, posed on dataset schemas, that result in textual visualization specifications that can be rendered into interactive data visualizations. Part of this framework includes applying LLMs to perform paraphrasing of natural language queries.
> 2. A dataset created with this framework for biomedical data portal metadata based on metadata from five major biomedical data portals.
>
> While DQVis was designed with training large models (including LLMs and VLMS) in mind, it is agnostic to a specific model, and we do not claim to make any contributions towards training models or evaluating existing models.
>
> **W4 Human Evaluation Clarification**
> The reviewer asks for clarification on the human evaluation process. We will answer the question here and point how how we plan to make the text in the paper more clear.
>
> *Recruiting*:
> We recruited five individuals with advanced degrees in computer science, data visualization, biomedical informatics, and professional experience in these domains. All individuals are familiar with HuBMAP and its metadata model. We will update Section 6 to make this information more apparent.
>
> *Stimuli*:
> We developed an interface that these participants used. This interface shows the question, the visualization, and some options for providing feedback on the data point. All five reviewers saw the data points for the first 20 questions, then random data points were sampled to get a greater diversity of questions reviewed. We include a screenshot of this interface in Figure 3e, but we will update Section 6.1 to reference this figure.
>
> *Results / Annotator Agreement*:
> The results for all 20 questions that all reviewers saw are reported in Figure 4d and the diversity of reviewer responses is discussed in Section 6.2. We will update the caption for Figure 4 and the text in Section 6.2 to further emphasize this. Additionally, for further transparency, we will include an additional figure or table in the supplemental material that includes all of the reviewer feedback for these 20 questions, as well as subcategories that reviewers included for the “bad” data points.
>
> We hope these clarifications are helpful and would be glad to provide further detail if needed.

---

### Official Review · Reviewer_GE5Y · 2025-07-01

**Rating:** 5
**Confidence:** 4

**Summary:**

This paper introduces DQVis, a new large-scale dataset for the natural language to visualization (NL2VIS) task, specifically tailored for the biomedical domain. The primary contributions are twofold: 1) a systematic, scalable framework for generating data-question-visualization triplets, and 2) the resulting DQVis dataset itself, which contains 1.08 million single-step query samples and 11.4 thousand two-step conversational samples.

**Dataset Code Accessibility:**

Yes

**Dataset Code Comments:**

The author has provided access links to the code and data.

**Ethical Considerations:**

No, there are no or only very minor ethics concerns

**Final Justification:**

I have no further concerns. Given the overall quality and comprehensive experiments, I will keep my score as 5.

**Limitations Weaknesses:**

1. As shown in Table 1, the dataset is heavily skewed towards the HuBMAP repository, which accounts for approximately 82% of the single-step triplets. The authors note this is because the HuBMAP data package contained more fields. However, this imbalance could cause models trained on DQVis to overfit to the specific schema, terminology, and data patterns of HuBMAP, potentially limiting their generalizability to other biomedical data portals.
2. The template-based approach, while scalable, has inherent limitations that are briefly mentioned in Section 7.1. It struggles to capture nuanced, variable-specific visualization conventions.
3. As mentioned by the authors in the limitations section, hallucination issues may arise when using large language models for text paraphrasing. The authors could supplement more test samples to more accurately estimate the probability of such hallucination phenomena, helping readers assess whether they might significantly impact the overall results. At the same time, I am quite curious whether the proportion of such hallucinations varies across different LLMs.

**Strengths Contributions:**

1. The paper is well-written, organized, and easy to understand. The logical flow from motivation to methodology to evaluation is clear. Figures and tables are highly informative.
2. The proposed DQVis dataset is both novel and significant. While general-purpose NL2VIS datasets like NVBench exist, DQVis is the first large-scale dataset specifically designed for the complex and high-impact domain of biomedical research data portals.
3. Another strength of this work is the commitment to quality control and evaluation. The authors did not simply generate data but also built a review interface and engaged five domain experts to assess the output

---

> ### Author Rebuttal · Authors · 2025-07-30
>
> ## Response:
>
> Thank you for your careful review of DQVis, your kind words with respect to the paper clarity, novelty, significance, and evaluation. Furthermore, thank you for your thoughts and feedback on how to make this paper even stronger.
>
> **Summary:**  The review points out several limitations with the dataset and approach, which are apparent in the paper. Given the score of 5-Accept, the contribution is still worthwhile with these limitations. We agree with this assessment and will further elaborate on why we believe the contributions remain valuable despite these three limitations, and how we will improve the manuscript to clarify them. Since Reviewer 4 (wTno) identified the same three limitations, some of the response text is repeated.
>
> **W1: Unbalanced Data Generation**
> DQVis includes data from five biomedical data repositories. The number of questions generated for each of these repositories is reported in Table 1, and the review points out that the majority of questions come from one of the five repositories (HuBMAP).
>
> We believe balancing the dataset during generation is not necessary and would be counterproductive since it would limit the applicability of the dataset. Instead, we recommend that users subsample or rebalance the dataset as appropriate for their task (e.g., by visualization type or dataset schema).
>
> Our approach is to generate as many questions as possible given our set of constraints and the supplied dataset schemas. This approach will generate an imbalance in the dataset due to differences in the dataset schemas, especially the number of entities and fields in the schema. We could subsample the space of possible valid questions to achieve more balance; however, this would reduce the number of generated data points.
>
> We will add a paragraph to the limitations section clarifying that the dataset is not balanced by design and recommending that users perform any desired rebalancing during model training or evaluation. We will provide guidance on how to accomplish this based on the provided templates and constraints.
>
> Finally, we emphasize that while this manuscript introduces the DQVis dataset, we are also proposing a novel dataset generation framework. This framework can easily incorporate additional dataset schemas. In the future, as additional, more diverse biomedical dataset schemas become available, they could be incorporated into the DQVis dataset or used to generate a new one.
>
> **W2 & W3: Template-Based Limitations & Paraphrasing Imperfections**
> The review recapitulates the limitations of template-based approaches and paraphrasing that we describe in Section 7.1 Limitations.
>
> Existing work in the domain illustrates the value of template-based approaches. For instance, an existing dataset paper for general natural language to visualization — *“Synthesizing Natural Language to Visualization (NL2VIS) Benchmarks from NL2SQL Benchmarks”,* Y. Luo, N. Tang, G. Li, C. Chai, W. Li, and X. Qin, in *Proceedings of the 2021 International Conference on Management of Data*, in SIGMOD ’21. doi: 10.1145/3448016.3457261 *— (NVBench),* transforms a natural language to a SQL dataset, which uses a template-based approach for generating data. Our approach goes beyond this existing work by applying modern LLMs for question paraphrasing.
>
> Paraphrasing with LLMs is currently the most effective and scalable method available, despite some known limitations. We adapt the approach proposed in — “Natural Language Dataset Generation Framework for Visualizations Powered by Large Language Models”, H.-K. Ko et al., in Proceedings of the 2024 CHI Conference on Human Factors in Computing Systems, in CHI ’24, doi: 10.1145/3613904.3642943 — which utilizes LLMs for paraphrasing natural language queries.
>
> We extend the approach in two key ways:
>
> 1. For paraphrasing of multi-step questions, our implementation provides detailed guidelines for the LLM, explicitly instructing it to preserve exact field names at higher formality/expertise levels and avoid unnecessary synonymization.
>
> 2. We also pass structured schema information alongside each paraphrasing task, enabling the LLM to refer to the dataset’s canonical field names and structure. This context supports accurate interpretation and reduces the risk of field name misrepresentation.
>
> The data we collected from five reviewers indicates that, the paraphraser’s mistakes do not dominate the dataset. Out of 357 reviewed examples, only 27 (7.6 %) were rated as bad (Figure 4b in our paper). Many of the problematic paraphrases stemmed from missing field descriptions (e.g. “rh_blood_group” lacked a definition in the metadata), which made it difficult for GPT‑4 to choose a meaningful synonym.
>
> It is worth noting that many natural language interfaces must handle user queries that are imprecise or even incorrect. Our dataset intentionally includes linguistic variety to help models handle such scenarios.
>
> In short, we argue that while templating and paraphrasing have limitations, they represent the current state-of-the-art approach, and we have take steps to mitigate the limitations and evaluate the results.

---

> > ### Comment · Reviewer_GE5Y · 2025-08-05
> >
> > Thank you for the response, I have no further concerns. I will keep my score.

---

> > ### Comment · Reviewer_GE5Y · 2025-08-09
> >
> > Regarding the details of the paper, I still have a few minor comments and questions:
> >
> > 1. In Figure 2, the numbering of the subfigures is incorrect; 'b' appears twice, and there is no subfigure 'e'.
> > 2. Why in the rewriting of multi-step queries is the selected score subset {1, 3, 5}, rather than a range from 1 to 5 as in single-step queries?
> > 3. In 5.2, the author define reifying as a "constraint satisfaction problem." For complex data schemas with a large number of entities and fields (such as HuBMAP), the number of entity-field combinations that need to be checked can become extremely large, leading to a "combinatorial explosion." Did the framework encounter performance bottlenecks when handling this issue? What is the computational cost of this instantiation step?
> > 4. Recent advancements in LLM-based multi-agent system have been rapid, and through the collaboration of agents in data analysis and programming, data visualization can be achieved by agents. This multi-agent-based approach can also overcome the limitations of predefined templates, offering greater flexibility and scalability. Does the author think that this multi-agent-based visualization approach will impact the future application prospects of this paper?

---

### Official Review · Reviewer_p7VV · 2025-07-08

**Rating:** 5
**Confidence:** 3

**Summary:**

The paper introduces a dataset for biomedical visualization generation from natural language queries. The dataset is built on top of 5 biomedical datasets, these datasets are paired with 1.08 million queries and their visualizations represented as a JSON specification. The dataset is also contains 11447 multi step exploration queries for a subset of the 1.08 single step queries. Additionally the dataset releases expert reviews of the queries. The dataset is generated through a process of generating templated questions, ensuring their correctness against a list of constraints, and paraphrasing the templated questions using LLMs. It seems that visualizations specifications are generated from the questions that adhere to the constraints.

**Additional Feedback:**

Please consider including the below details in the paper.
- Sec 5.1: How many question templates are used to generate the dataset? What is the distribution of the question templates in the ten task types? (Lines 187-188)
- Sec 5.2: How many constraints is a question evaluated against? Are the constraints question specific? Is it trivial to obtain the questions which satisfy all the constraints? e.g. can each question be exhaustively checked against the constraints? If not, please elaborate.
- Sec 5.3: What exactly is technicality and formality? Please include the prompt used to get these scores.
- Sec 5.4: "We then designed 17 templates links. These link broad questions to more specified ones — a common step when performing exploratory data analysis...." - this is a reasonable insight, but please describe the basis for this? Was it based on analysis of realistic exploratory sessions? Was it based on prior work, etc?
- Please consider citing and discussing: https://dl.acm.org/doi/10.1145/3637528.3671935

**Dataset Code Accessibility:**

Yes

**Ethical Considerations:**

No, there are no or only very minor ethics concerns

**Final Justification:**

The authors response is sufficient and im happy to see this paper accepted.

**Limitations Weaknesses:**

- Some details of the dataset construction could be better described.

**Strengths Contributions:**

- The paper contributes a large scale dataset for a realistic and impactful task.
- The papers dataset construction procedure seems sound and well described.

---

> ### Author Rebuttal · Authors · 2025-07-30
>
> Thank you for your time and effort reviewing DQVis, positive comments regarding the scale of our dataset and soundness of our approach and description, and detailed suggestions for improving this manuscript.
>
> **Summary:** The review asks for additional clarifying information to improve the communication of the work. These changes are all easy to include in a revision and will make the final paper even clearer.
>
> **Clarification 1: Sec 5.1: How many template questions are there, and what is the distribution across task types?**
> The dataset contains 64 template questions. We will include this information either in text or as a table summarizing the counts across different analytical task types. The table will either be included in the paper or supplemental material, depending on space constraints.
>
> **Clarification 2: Sec 5.2: How many constraints is a question evaluated against? Are the constraints question specific? Is it trivial to obtain the questions which satisfy all the constraints?**
> Constraints are question-specific, though some constraints are applied to multiple questions. There are between 1 and 11 constraints for each question. We do generate every possible question that satisfies all of the constraints. This is accomplished with an existing constraint satisfaction solver library. So with this technical setup in place, it becomes straightforward to generate all valid questions. We will update the text in section 5 to include these clarifications.
>
> **Clarification 3: Sec 5.3: When paraphrasing natural language queries, how is technicality and formality defined?**
> We define the dimensions of technicality and formality based on previous work that we already cite in the paper (“Natural Language Dataset Generation Framework for Visualizations Powered by Large Language Models”, H.-K. Ko et al., in Proceedings of the 2024 CHI Conference on Human Factors in Computing Systems, in CHI ’24, doi: 10.1145/3613904.3642943). We will update the text to clarify that we are using their definitions for these dimensions and will include the full prompts for paraphrasing in the supplemental material.
>
> **Clarification 4: Sec 5.4: When designing multi-step question links, what is the basis for linking broad questions to more specific ones?**
> When designing the logic to link follow-up questions, we considered models of the information-seeking process. Several models are summarized in Chapter 3 of Search User Interfaces (Hearst, 2009). One model proposed by Shneiderman et al. 1997 includes four steps. 1\. Query Formulation, 2\. Action, 3\. Review Results, 4\. Refinement. The other models have slightly different formulations, but all agree on an interactive cycle where results are acquired and actions are refined based on those results. So in our case, the multi-step questions include an initial question (steps 1 and 2), a resulting visualization (step 3). Then, based on the information presented, a refined follow-up query that requests additional information (Step 4). We will include more of this context in the paper.
>
> **Regarding the paper with doi: 10.1145/3637528.3671935**
> Thank you for pointing us to this publication. We will include this work in our related work discussion.

---

### Note · Authors · 2025-08-14

Dear NeurIPS 2025 Reviewers, AC, SAC, and PC,

We sincerely appreciate your time and effort reviewing our paper.

In this work we contribute a new dataset (DQVis) consisting of natural language queries about biomedical data portal metadata with corresponding visualization specifications.

Reviewers highlighted several strengths of our work:
- **Domain-specific impact**: DQVis addresses the impactful domain of biomedical research (*p7VV*, *GE5Y*).
- **Expert review**: The dataset was assessed by domain experts (*GE5Y*, *Zn66*).
- **Multi-step reasoning support**: Includes 11.4k two-step query pairs (*Zn66, *wTno*).
- **Clarity**: The paper is well-structured and clearly communicated (*p7VV*, *GE5Y*).

We thank the reviewers for their detailed comments. All concerns were addressed in our rebuttal, and we will incorporate clarifications into the manuscript. Two key points include:

- **Template Based Generation + Paraphrasing**: This approach is the current state of the art, we take steps to mitigate limitations, and our review shows this is a minor problem.
- **Dataset Balance**: We will include additional details in the paper on the current dataset balance and provide provide guidance on balancing data in different ways.

We note that the concerns raised by reviewer *wTno* overlap closely with those from **reviewer *GE5Y*, who agreed during the discussion that our rebuttal sufficiently addressed them. We did not received any any responses or Mandatory Acknowledgement from Reviewer *wTno*,** so their score was not informed by our clarifications.

Finally, we appreciate reviewer *GE5Y*’s additional suggestions beyond the discussion phase. While there was no opportunity to respond within the formal process, we will incorporate these minor improvements into our revisions.

Best,
Authors

---

### Decision · Program_Chairs · 2025-09-18

**Decision:**

Accept (poster)

**Comment:**

Summary:

The paper discusses creation of a data-question visualization generation
framework and DQVis - a dataset constructed with the framework for the
biomedical domain (specifically for 5 biomedical datasets).

Strength:

- Evaluated by five experts
- Important area


Weakness:

- Experts	rate some data bad and experts were not very consistent
with each other.  While authors report that only 7.6% of
the overall questions were marked bad, it is clear from the 20 data
points reviewed by all reviewers, that many reviewers are missing bad
aspects.  From the 20 data points reviewed by all reviewers, 13 or 20
were found bad by at least one reviewer.

- Use of GPT-4o for paraphrasing has some	problematic mismatches

- Template-driven	semantics is not fully flexible.



Decision Reason:

It is a tough call.  Reviewers were generally positive (4,4,5,5) while
recognizing the same limitations mentioned by the authors.  I am most
concerned about the quality of the dataset, specifically that from the
20 data points by all reviewers, 13 or 20 were found bad by at least
one reviewer.

Discussion:

Discussion clarified that reviewers and	the authors agree on the
limitations of the work, but mostly feel that the work is worth
publishing.